# Dependency of B-Cell Acute Lymphoblastic Leukemia and Multiple Myeloma Cell Lines on MEN1 Extends beyond MEN1–KMT2A Interaction

**DOI:** 10.3390/ijms242216472

**Published:** 2023-11-17

**Authors:** Tatjana Magdalena Wolffhardt, Franz Ketzer, Stefano Telese, Thomas Wirth, Alexey Ushmorov

**Affiliations:** 1Institute of Physiological Chemistry, University of Ulm, Albert-Einstein-Allee 11, 89081 Ulm, Germany; tatjana.wolffhardt@uni-ulm.de (T.M.W.); telese97@gmail.com (S.T.); 2Center for Molecular and Cellular Oncology, Yale School of Medicine, New Haven, CT 06510, USA; franz.ketzer@yale.edu

**Keywords:** menin/MEN1, targeted therapy, B-cell acute lymphoblastic leukemia (B-ALL), multiple myeloma (MM)

## Abstract

Menin/MEN1 is a scaffold protein that participates in proliferation, regulation of gene transcription, DNA damage repair, and signal transduction. In hematological malignancies harboring the KMT2A/MLL1 (MLLr) chromosomal rearrangements, the interaction of the oncogenic fusion protein MLLr with MEN1 has been shown to be essential. MEN1 binders inhibiting the MEN1 and KMT2A interaction have been shown to be effective against MLLr AML and B-ALL in experimental models and clinical studies. We hypothesized that in addition to the MEN1–KMT2A interaction, alternative mechanisms might be instrumental in the MEN1 dependency of leukemia. We first mined and analyzed data from publicly available gene expression databases, finding that the dependency of B-ALL cell lines on MEN1 did not correlate with the presence of MLLr. Using shRNA-mediated knockdown, we found that all tested B-ALL cell lines were sensitive to MEN1 depletion, independent of the underlying driver mutations. Most multiple myeloma cell lines that did not harbor MLLr were also sensitive to the genetic depletion of MEN1. We conclude that the oncogenic role of MEN1 is not limited to the interaction with KMT2A. Our results suggest that targeted degradation of MEN1 or the development of binders that induce global changes in the MEN1 protein structure may be more efficient than the inhibition of individual MEN1 protein interactions.

## 1. Introduction

Mutated in endocrine neoplasia (menin/MEN1) is a scaffold protein that participates in multiple biological processes, including proliferation, regulation of gene expression, and DNA damage repair [1]. MEN1 is ubiquitously expressed and can either act as a tumor suppressor or facilitate tumor growth in a tissue-specific manner. In endocrine organs, MEN1 represses the transcription of cyclin-dependent kinase inhibitors p18/CDNK2C and p27/CDKN1B, and its inactivation by oncogenic mutations predisposes to malignant transformation [1]. Interestingly, hematological malignancies harboring chromosomal rearrangements of the epigenetic effector histone methyltransferase KTM2A/MLL1 (MLLr) become addicted to MEN1 expression [2,3]. MLLr results in the expression of oncogenic fusion proteins consisting of the N-terminus of KMT2A fused to the C-terminus of one of over 80 different partners. The MLLr fusion proteins enhance the transcription of genes, activating the proliferation of hematopoietic cells, including *HOXA* genes, *MEIS1* [4], *CDK6*, *MEF2C* [5], and *FLT3*, that might have major pathogenetic significance [2]. MEN1 lacks enzymatic or DNA-binding activity; instead, it serves as a link between transcription factors and epigenetic effectors like KMT2A [1]. The central cavity of MEN1 contains a binding site for KMT2A and is highly suitable for targeting by small molecules to inhibit MEN1–KMT2A interactions [6]. Consequently, a variety of MEN1 binders have been designed and have demonstrated high efficacy against MLLr B-ALL and AML in experimental models [5,7]. Most MEN1 binders were highly selective towards MLLr cell lines in comparison to MLLr-negative leukemias [5]. In addition, nucleophosmin (NPM1m) [8] and meningioma-1 mutations (MN1mut) [9] confer dependency on MEN1 and sensitivity to inhibitors of MEN1–KMT2A interaction despite the non-rearranged *KMT2A* in these neoplasms [10]. Clinical studies confirmed an antitumor effect of KO-539 and VTP50469/SNDX-5613 against MLLr and NPM1m MLL and B-ALL, which had previously been observed in pre-clinical experiments [11,12]. In particular, SNDX-5613 conferred complete remission (CR) in 44% of patients with a median duration of response of 5.4 months in MLLr AML and ALL, and in mNMP1 AML [13], while KO-539 induced CR in 2 of 12 MLLr or NMP1m AML patients [14].

Surprisingly, a dependency on MEN1 was revealed not only in MLLr-, NPM1m-, or MN1m-driven leukemia, but also in other types of neoplasms (Ewing sarcoma, breast cancer, and castration-resistant prostate cancer), which were driven by alternative mutations [15,16,17]. Given that MEN1 participates in multiple protein–protein interactions (PPI), thereby regulating various pro-survival mechanisms, we hypothesized that mechanisms other than MEN1–KMT2A PPI are instrumental in MEN1 dependency.

To test this hypothesis, we analyzed publicly available data from genome-wide loss-of-function screenings. Comparing AML and B-ALL, two leukemia types often driven by MLLr, MEN1 dependency correlated with the presence of MLLr only in AML. We validated the data of CRISPR loss-of-function screenings using shRNA-dependent knockdown and found that all tested B-ALL cell lines were sensitive to MEN1 depletion, independently of the type of driver mutations. Vice versa, only cell lines driven by MLLr were sensitive to VTP50469. Most of the multiple myeloma (MM) cell lines were also sensitive to the genetic depletion of MEN1, but none of them were sensitive to the inhibitor of MEN1–KMT2A interaction VTP50469 at therapeutically relevant doses. We conclude that the role of MEN1 in leukemia is not limited to its interaction with KMT2A, hence pharmacological degradation may be a better approach than the inhibition of individual MEN1 PPIs.

## 2. Results

### 2.1. MEN1 Dependency Is not Unique to Neoplasms Harboring MLLr, NPM1m, or MN1m

Previous analyses of genome-wide loss-of-function screening datasets of the Achilles project revealed the highest dependency on MEN1 in “multiple myeloma”, followed by “leukemia” categories [18]. The loss-of-function screening datasets are constantly updated and have been harmonized with genomics and cellular model data within the Cancer Dependency Map (DepMap) project (https://depmap.org/portal/depmap/; accessed on 24 October 2022). Thus, we used DepMap to corroborate previous data (Figure 1A).

In this analysis, seven of eight malignancies with the highest dependency on MEN1 belong to the group of hematologic malignancies. “Plasma cell myeloma” and “B-lymphoblastic” categories ranked 4th and 8th, respectively. In this respect, our updated analysis correlates with previously published data. To clarify whether the presence of MLLr is a prerequisite for MEN1 dependency, we compared MEN1 dependency in individual AML and B-ALL cell lines. MLLr AML cell lines were expectedly most dependent on MEN1 expression (Figure 1B,D). Intriguingly, this did not hold true for B-ALL (Figure 1C). Finally, we analyzed the dependency data of individual MM cell lines, none of which carry MLLr, NPM1m, or MN1m. In 5 out of 21 MM cell lines, the calculated gene effect was lower than -1 (mean value of the canonical oncogenes), indicating frequent dependency of MM on MEN1 (Figure 1E).

To further understand the role of MEN1 in the oncogenic programs of AML, B-ALL, and MM we correlated the effect of MEN1 expression to the overall survival of patients (Figure 1F). The analysis revealed significantly worse survival in MEN1^high^ B-ALL in the TARGET Phase 1 study of precursor B-ALL, while no significant effect was observed in the ALL of ambiguous lineage (TARGET-ALL-P3).

In MM (MMRF-COMPASS study), the overall survival of patients with higher MEN1 expression was significantly worse over the course of the study. In patients with MEN1^high^ in particular, high MEN1 expression is significantly adversely correlated with the efficacy of the Total Therapy 6 (TT6) treatment protocol consisting of induction therapy with Melphalan/Bortezomib/Thalidomide/Dexamethasone/Cisplatin/Doxorubicin/Cyclophosphamide/Etoposide (M-VTD-PACE) followed by a high-dose M-VTD-PACE-based tandem transplant [19], and in some molecular subtypes of MM [20] (Appendix A).

AML did not show significantly decreased overall survival (Figure 1F). The lack of a significant effect of high MEN1 expression on patient survival in AML was surprising, considering several recent publications have shown that MEN1 is a molecular dependency in various subtypes of AML [5,21,22,23]. However, when analyzing molecular subtypes within the AML study cohort, several AML-subtypes show worse prognosis with high MEN1 expression, including NPM1-mutated, which has been shown to be sensitive to MEN1 inhibition [22,23] (Appendix A).

Based on our analysis of MM and AML subtypes, it is important to understand the relationship between MEN1 expression and the effects of MEN1 inhibition or degradation.

### 2.2. MEN1 Knockdown Decreases Fitness and Induces Apoptosis in B-ALL Cell Lines Independently of Mutational Background

To corroborate the data of these loss-of-function screenings, we used shRNA-dependent knockdown of *MEN1* in four B-ALL cell lines: NALM6 (ETV6-PDGFRB), SUP-B15 (BCR-ABL1), and the MLLr cell lines RS4;11 (MLL-AF4) (Figure 2) and KOPN8 (MLL-ENL) (Figure 3). Both shRNAs similarly decreased MEN1 protein expression in all cell lines (Figure 2A) and compromised cell fitness in a competitive growth assay (Figure 2B).

The decrease in fitness in cells transduced with the MEN1-targeting shRNAs was associated with the induction of apoptosis, which was stronger in MLLr-negative cell lines (Figure 2C). Notably, in the MLLr B-ALL cell lines RS4;11 (Figure 2B,C) and KOPN8 (Figure 3B,C) the MEN1 knockdown induced less cell death than in the NALM6 and SUP-B15 cell lines driven by other oncogenic mutations. At the same time, there were no visible differences between the cell lines with respect to a decrease in fitness, indicating that other mechanisms, e.g., cell cycle arrest, might be instrumental.

To exclude a dependency of the NALM6 and SUP-B15 cell lines on the MEN1–KMT2A interaction, we measured their sensitivity to the specific inhibitor VTP50469. Both cell lines were resistant to VTP50469 (IC_50_ > 20 µM), whereas the IC_50_ values of the MLLr cell lines RS4;11 and KOPN8 were 35 nM and 15 nM, respectively (Figure 2D and Figure 3D).

Thus, shRNA-dependent knockdown corroborated the data mined from the genome-wide loss-of-function CRISPR screenings, showing universal dependence of B-ALL on MEN1 expression.

### 2.3. MEN1 Knockdown Decreases Fitness and Induces Apoptosis in MM Cell Lines

We also used the shRNA-dependent approach to investigate the effect of MEN1 knockdown in MM cell lines. Both shRNAs decreased MEN1 protein expression in all MM cell lines (Figure 3A) and induced a significant decrease in cell fitness in a competitive growth assay, albeit to different extents (Figure 3B).

MEN1 knockdown induced apoptosis in all MM cell lines, albeit to a different extent (Figure 3C). Next, we assessed the sensitivity of five MM cell lines to VTP50469. Neither of the cell lines was sensitive to the inhibitor (IC_50_ > 10 µM) (Figure 3D). To address the role of KMT2A in cell death induced by MEN1 downregulation, we measured the effect of MEN1 KD on KMT2A protein expression. Surprisingly, in two of three MM cell lines, the KD of MEN1 was associated with a decrease in KMT2A at the protein level (Figure 3E).

Thus, MEN1 depletion but not the inhibition of MEN1–KMT2A interaction induces apoptosis in MM cell lines. Importantly, MEN1 depletion might downregulate KMT2A protein expression in some subtypes of MM.

### 2.4. CRISPR/Cas9-Dependent Inactivation of MEN1 Decreases the Fitness of MM and B-ALL Cells Harboring Edited Alleles

The observed cytotoxic effect of MEN1 KD in the cell lines that do not harbor MLLr or other known mutations that endow sensitivity to the MEN1 binders was too surprising to be accepted without additional proof. Neither the RNA interference, which we used for MEN1 KD nor CRISR/Cas9-dependent DNA editing, which was used in gene-wide loss-of-function screening, which we used in our analyses, are free of off-target effects, potentially confounding the interpretation of the results [24,25]. In addition, the lentiviral transduction which was used to deliver Cas9 and gRNA constructs in the whole genome screening itself induces pro-apoptotic JUN and deregulation of miRNA transcription [26], confounding the interpretation of their results. Moreover, stable expression of Cas9/gRNA by lentiviral vectors increased the probability of off-target DNA editing [27].

Given that CRISPR editing is prone to off-target effects [28], we chose this method for further optimization. We took advantage of the nucleofection of the ribonucleoprotein (RNP) complexes consisting of Cas9 and two tandem crRNA:tracrRNAs which allows a “hit-and-run” genome editing of the first coding exon of MEN1, and reduces the risks of continued off-target mutagenesis and increases on-target efficiency [27] (Figure 4A).

To analyze the effect of the MEN1 KO on cell fitness, we monitored the dynamics of knockout alleles in the whole cell population. To this end, we calculated the knockout score (percentage of edits that result in a frameshift or an indel more than 21 bp in length) at different time points after transfection. We preferred this dynamic approach to the end-point measurement (as in the case of CRISPR/Cas9 genome-wide loss-of-function screenings) and to the selection of single KO clones due to the inherent tendency of CRISPR/Cas9 DNA editing to induce TP53-dependent DNA damage response followed by growth arrest and apoptosis [29]. Given the intrinsic heterogeneity of tumor populations [30], this ultimately leads to the selection of TP53-deficient clones that are resistant to apoptosis and have a higher proliferation potential [31]. Thus, the selected CRISPR KO clones might not represent the naïve cell population, and the results of the CRISPR loss-of-function screenings can be confounded by the expansion of the TP53-deficient populations [25,29]. To control for cytotoxic effects of the CRISPR/Cas9-induced DNA damage response on the growth and survival parameters of the cells we targeted the third intron of troponin I3, cardiac type (TNNI3), which is not essential in B-ALL and MM cell lines (DepMap, accessed on 13 November 2022).

Transfection of the RNP complex targeting the first coding exon of MEN1 (Figure 4A) efficiently reduced MEN1 protein expression (Figure 4B) and resulted in the continuous decrease in the proportion of the KO MEN1 alleles in all cell lines (Figure 4C). In the MLLr cell line RS4;11, the proportion of the MEN1 KO alleles decreased more quickly and became undetectable at day 14. In the non-MLLr B-ALL cell line NALM6 and the MM cell line L363, the population of cells carrying edited MEN1 alleles became smaller over the course of 14 days in a steadily progressive manner. The proportion of cells carrying edited TNNI3 control alleles remained the same in all cell lines (Figure 4C).

We concluded that MEN1 depletion confers a selective disadvantage in B-ALL and MM cell lines which do not harbor MLLr, or other mutations conferring dependency on MEN1.

## 3. Discussion

In this proof-of-principle study, we addressed the feasibility of MEN1 depletion vs. specific targeting of the MEN1–KMT2A interaction for the treatment of lymphoid malignancies, including those that are not driven by MLLr and other known mutations endowing sensitivity to the inhibitors. We found that in contrast to the inhibition of the MEN1–KMT2A interaction, the depletion of the MEN1 protein is toxic for B-ALL cell lines independently of the driver mutations. Moreover, we corroborated the data of genome-wide loss-of-function screenings on the dependency of MM cell lines on MEN1 and showed that in some cases the depletion of MEN1 is associated with a decrease in KMT2A expression on the protein level.

We have proven that cell lines that harbor neither MLLr nor other mutations conferring sensitivity to the inhibitors are dependent on MEN1 expression. This is in line with the reported high selectivity of VTP50469 [5] and other MEN1 inhibitors, including MI-3454 [32], towards MLLr leukemia. We concluded that dependency on MEN1 cannot be explained exclusively by its interaction with KMT2A. MEN1 participates in multiple PPIs and it is conceivable that inactivation of only one of them might not suffice to compromise cell fitness. MEN1 interacting with different proteins contributes to multiple pro-survival pathways, including DNA replication and repair, regulation of the activity of transcription factors, control of protein stability, and regulation of signaling pathways. In particular, MEN1 interacts with replication protein A2 (RPA2), which is involved in DNA replication, proliferation, and repair [33], and FANCD2, a protein mutated in patients with an inherited cancer predisposition syndrome [34]. MEN1 mutations disturbing interaction with these proteins compromise DNA repair, thereby facilitating oncogenesis. It is conceivable that depletion of MEN1 might increase the sensitivity of the tumor cells to DNA-damaging therapeutics.

MEN1 is involved in the assembly of multiple transcription-regulating complexes, and can both activate or repress transcription recruiting either epigenetic activators or repressors [35]. MEN1 binds to and represses the transcriptional activity of potentially oncogenic transcription factors like JUND [36] and NF-κB [37]. Moreover, MEN1 increases the stability of the transcription factor FOXO1 [38], which might promote or repress tumor growth in a dose-dependent manner [39,40]. Interestingly, MEN1 also inhibits the activity of PI3K-PDPK1-AKT signaling, which activates cell proliferation and metabolism, i.e., by induction of FOXO1 degradation [41]. Although deemed as a pro-survival pathway, AKT activity is tightly regulated in B-ALL and both “too low” and “too high” activity are detrimental to pre-B-cells and B-ALL [42]. Similarly, we demonstrated the importance of tight regulation of AKT activity in BL [43]. It is therefore conceivable that MEN1 contributes to tumor survival by maintaining the activity of essential transcription factors and oncogenic signaling at a “just right” level in terms of the Goldilocks paradigm [44]. Therefore, depletion of MEN1 might show a superior antitumor effect compared to the inhibitors of MEN1–KMT2A interaction, specifically in B-cell-derived malignancies which depend on tightly controlled signaling.

Our data on the dependency of MM on MEN1 are of specific interest because MLLr and related mutations are not common in MM [45]. Our data demonstrating the sensitivity of MM to loss of MEN1 are in line with previous analyses of loss-of-function screenings, where MM was ranked as the most MEN1-dependent tumor entity [18]. In our analyses, all tested MM cell lines were resistant to the inhibitor of MEN1–KTM2A interaction VTP50469, indicating that MEN1–KMT2A PPI does not play a major role in the dependency of MM on MEN1. Curiously, our data on the decrease in KMT2A protein expression after MEN1 depletion do not allow us to exclude a MEN1-independent role of KMT2A in the oncogenic program of MM. Although not present in all tested cell lines, the concurrent downregulation of KMT2A after MEN1 knockdown could be indicative of a potential role of MEN1 in the transcription or posttranslational stability of the KMT2A protein. Additionally, our results hint at a previously unknown role and dependency of some non-MLLr leukemias on KMT2A expression. Moreover, the dependency of KMT2A on MEN1 expression is not universal among different tissue types. For instance, MEN1 KO in the myelogenous cell line K562 did not decrease the KMT2A protein [46]. Nevertheless, KMT2A downregulation is not the only mechanism of MEN1-KO-induced toxicity in B-ALL and MM, since it was not observed in all cell lines.

Importantly, selective inhibition of the MEN1–KMT2A or MEN1–KMT2A–FP interaction is considered a guarantee of the low toxicity of the inhibitors [6]. On the other hand, the therapeutic spectrum of highly specific inhibitors like VTP50469 appeared to be narrow and included only neoplasms driven by MLLr, NPM1m, or MN1m [5]. Interestingly, other inhibitors of MEN1–KMT2A might have a broader spectrum of antitumor activity. For example, Ewing sarcoma cell lines appeared to be sensitive to the non-covalent inhibitor of MEN1–KMT2A interaction MI-503 [15], but not to VTP50469 [47]. In addition, MI-503 inhibited growth in breast and prostate cancer cell lines. At the same time, a large-scale study of the cytotoxic spectrum of BAY-155, structurally similar to MI-503, identified a few sensitive cell lines in different types of neoplasms, including multiple myeloma, colorectal cancer, as well as tumors of the esophagus, liver, and uterus [18]. Notably, BAY-155 and MI-503 decreased MEN1 protein expression in a tumor- and concentration-dependent manner [15,18]. In Ewing sarcoma cell lines, the decrease in protein expression was even associated with an antitumor effect [15]. It is conceivable that the effect of BAY-155 and MI-503 might be explained not only by off-target activity but also by induction of unexpected conformational changes in the MEN1 protein. The conformational changes might increase the sensitivity of MEN1 to protein degradation, or interfere with MEN1 PPIs, in a similar way as the binding of rapamycin to FKBP12 facilitates its inhibitory binding to mTOR [48].

Most recently, preliminary data on the antitumor effect of the new irreversible covalent MEN1 binder BMF-219 [49] have been published. BMF-219 appeared to be toxic not only to the AML and B-ALL cell lines harboring MLLr, NPM1m, or MN1m, but also to several other cell lines including chronic lymphocytic leukemia CLL, DLBCL, and MM [50,51]. It is conceivable that the irreversible binders might cause conformational changes in the MEN1 molecule and thereby influence multiple MEN1 PPIs. Alternatively, they may facilitate the formation of neo-PPIs of MEN1 [52], which ultimately induce cell death. We anticipate that the design of MEN1 binders that induce global structural changes in MEN1 will yield more efficient compounds than non-covalent inhibitors like VTP50469.

Our study also points to protein degradation as a prospective approach to developing new drugs targeting MEN1. The stability of proteins can be selectively targeted by proteolysis targeting chimeras (PROTACs), a new group of drugs, which allow selective degradation of proteins by targeting them to the ubiquitin-proteasomal system. PROTACs consist of two ligands joined by a linker. One ligand, the “warhead”, binds a protein of interest, while the other recruits an E3 ubiquitin ligase which transfers ubiquitin to the protein of interest to facilitate its proteolytic degradation [53]. Given that the MEN1 binders can be used as “warheads”, the design of the PROTACs against MEN1 is feasible.

Conclusively, our study provides the rationale for targeting MEN1 for degradation, for example by PROTACs. According to our data, this approach would be more efficient and have a broader antitumor spectrum than specific inhibitors of MEN1–KMT2A interaction. It is conceivable that changing the paradigm of the design of MEN1 binders towards compounds that remodel physiological MEN1 PPIs or facilitate the building of neo-PPIs will reveal a new, hitherto unknown potential of MEN1 targeting.

## 4. Materials and Methods

Additional and detailed information on all experimental procedures and reagents is provided in the Appendix A.

### 4.1. Cell lines and Culture Conditions

B-ALL cell lines NALM6, RS4;11, SUP-B15, were cultured in complete RPMI1640 medium containing 20% of FCS (Gibco, Waltham, MA, USA), whereas KOPN-8 in complete medium containing 10% FCS. MM cell lines U266; OPM2, and MM1.s were cultured in RPMI medium containing 10%, for KMS-12-BM 20%, for L363 15%. MM LP1 cells were cultured in IMDM medium (PAN-Biotech, Aidenbach, Germany) containing 20% FCS. The HEK293T cells were cultured in complete DMEM medium (Gibco) con-taining 10% FCS. In addition to FCS complete medium was always complemented with 2 mM L-glutamine, 100 U/mL penicillin, 100 µg/mL streptomycin, and 50 µM monothioglycerol for the MM cell lines. All cells were cultured at 37 °C and 5% CO_2_.

### 4.2. shRNA-Mediated Knockdown of MEN1

The shRNA target sequences for MEN1 were selected from Broad Institute RNAi consortium shRNA library (https://www.broadinstitute.org/rnai-consortium/rnai-consortium-shrna-library, accessed on 2 March 2022). Corresponding oligonucleotides were assembled and cloned into pRSI12-U6-sh-UbiC-TagRFP-2A-Puro vector (Cellecta, Logue Ave, CA, USA). ShRNA target sequence as well as amplification primers are listed in the Appendix A (Appendix A) Lentiviral transduction was done as described previously [39].

### 4.3. CRISPR/Cas9 Gene Editing and Analysis of the Dynamics of CRISPR-Induced InDels

Tandem CRISPR/Cas9-dependent gene inactivation was performed similarly as originally described [31] with some modification as described in the Appendix A. crRNA sequences are listed in Appendix A. The crRNAs targeting 2nd exon of *MEN1* and control crRNAs complementary to the 3rd *TNNI3* intron were selected with help of Custom Alt-R™ CRISPR-Cas9 guide RNA on-line tool (https://eu.idtdna.com/site/order/designtool/index/CRISPR_CUSTOM; accessed on 9 June 2022). The crRNAs, tracrRNA, and Alt-R^®^ S.p. Cas9 Nuclease V3 were obtained from IDT (Leuven, Belgium). To assemble RNP complexes 200 µM of each tandem crRNA and 200 µM tracrRNA were mixed at equimolar ratio, heated at 95 °C 2 min and cooled down at room temperature. Once the mix has cooled, 450 pmol of each of the two different crRNAs:tracrRNA complexes were added to 150 pmol in total of Cas9 per nucleofection. The RNP mix was incubated in the dark at room temperature for 15min. In the meantime, 5 × 106 cells werte washed once in PBS and centrifuged at 1200 rpm for 5 min. The cell pellet was resuspended in 100 µL of corresponding nucleofection buffer and transfected with help of 4D-Nucleofector^®^ X- (Lonza, Basel, Switzerland). Typically, 5 × 106 cells were nucleofected with RNP complexes using 100 µL cuvettes. Nucleofected cells were incubated in complete medium at normal conditions. Medium was changed every three days to maintain cell density of 0.5–1.0 × 106 per ml. Starting from the day 4 the cells were harvested with 7 day intervals. gDNA was extracted from up to 10 × 106 cells with help of DNeasy Blood & Tissue kit (Qiagen, Hilden, Germany). To amplify edited regions, we used PCR primers targeting flanking sequences. The primers were synthesized by Biomeres.net GmbH (Ulm, Germany). The gDNA was amplified with help of GoTaq DNA Polymerase (Promega, Madison, WI, USA) in the presence of 5% DMSO and the amplified fragments we visualized on 1% agarose gel and purified with help of QIAquick gel extraction kit (Qiagen). The DNA fragments were sequenced by Eurofins (Eurofins Scientific SE, Luxembourg City, Luxembourg). To assess the spectrum of the InDels, the Sanger chromatograms were analyzed with help of Inference of CRISPR Edits (ICE) CRISPR Analysis tool (https://ice.synthego.com, accessed on 10 June 2022) Synthego Corporation (Redwood City, CA, USA). This method confer analysis of the indels, generated by CRISPR-editing at the resolution level of next generation sequencing. To assess dynamics of potentially inactivating indels we used ICE parameter “Knockout Score”, which represents the proportion of cells that have either a frameshift or indel exceeding 21 bp. To control potentially pro-apoptotic effect of CRISPR-induced double strand break or to exclude selection of TP53-deficient clones, which is an intrinsic setback of CRISPR-dependent DNA editing, as control we edited 3rd intron of *TNNI3*.

### 4.4. Immunoblotting

Immunoblotting was done as we described previously [40]. Primers used for qRT-PCR and antibodies used for immunoblotting are listed in Appendix A. For immunoblotting the cells were lysed in SDS sample buffer supplemented with protease inhibitor cocktail tablet (Roche Diagnostics International AG, Basel, Switzerland)) and boiled for 5–10 min at desaturating conditions. The proteins were separated by SDS-PAGE and transferred to a nitrocellulose membrane (Amersham Pharmacia, Piscataway, NJ, USA). After incubation with antibodies the protein bands were visualized by addition of SuperSignal West Dura Extended Duration Substrate (Thermo Fisher, Waltham, MA, USA).

### 4.5. Assessment of Apoptosis, Cell Viability, and Competitive Co-Culture Assay

Apoptosis, cell viability, and determination of IC_50_ IC50 by MTT test was done over the course of five days as we described previously [43]. To assess apoptosis the cells were stained with Annexin-V-FITC (BD Biosciences) and Propidium Iodide (Sigma-Aldrich, St. Lois, MO, USA). The cell death was measured with help of FACS Canto II, and analyzed with FlowJo software (BD Biosciences). To measure cell viability, we used metabolic MTT assay. Cells were seeded into 96 well plates at a density of 2 × 10^4^ cells per well in complete medium. Solvent control wells were incubated with dimethyl sulfoxide (DMSO). Positive control wells were treated with 5 µg/mL puromycin (#540222, Merck, Darmstadt, Germany). Cells were incubated at standard conditions for 5–6 days, followed by addition of 25 µL of the 5 mg/mL MTT solution (Sigma-Aldrich) for 2 h at normal culture conditions. Then, 100 µL lysis buffer (20% SDS, 50% dimethylformamide, 2% acetic acid, 0.15 mM HCl, pH 4.7) was added and after an overnight incubation at 37 °C, the optical densities (OD) were measured at 570 nm wavelength using the SpectraMax 250 microplate reader (Molecular Devices, San Jose, CA, USA) with help of the SoftMax Pro 3.0 software (Molecular Devices; RRID: SCR_014240). Percentage of growth inhibition at a given drug concentration was calculated as (1 − OD_drug_ − OD_puromycin_)/OD_DMSO_ × 100. The half maximal inhibitory concentration was calculated using GraphPad Prism software (RRID: SCR_002798). For competitive growth assay percentages of RFP^+^ cell populations were analyzed using the BD FACSDiva Software (RRID:SCR_001456) or Flow Jo Software (Version 10.8.1, RRID: SCR_008520). The percentage of RFP^+^ cells was measured every 3 days. First measurement was performed 4–5 days post transduction and the percentage of transduced cells was set as 100% and used for normalization of results of following measurements.

### 4.6. Statistical Analysis

CRISPR dependency data for MEN1 were mined from DepMAP (DepMap 22Q4, Chronos, https://depmap.org, accessed on 28 February 2023) database which comprises data on genome-wide loss-of-function CRISPR/Cas9 screenings. For the analysis we included all AML, B-ALL, and MM cell lines available in the database at that time of analysis. DepMap calculates “gene effect scores” to measure the effect of knocking out of the gene on the viability of a given cell line. These values are calculated using the Chronos algorithms [54]. The data are normalized using annotated sets of common-essential and non-essential genes, such that a score of 0 represents no viability effect, and a score of −1 corresponds to the median effect of known common-essential genes. Correspondingly, negative scores represent cell lines that are most dependent on the gene. CRISPR dependency scores for MEN1 were mined from DepMAP (DepMap 22Q4, Chronos, https://depmap.org, 6 August 2022). The calculation of IC_50_, and statistical analyses of the data using Mann-Whitney-Test and regression analysis were done with GraphPad PRISM (GraphPad Software, San Diego, CA, USA; RRID:SCR_002798).

## Figures and Tables

**Figure 1 ijms-24-16472-f001:**
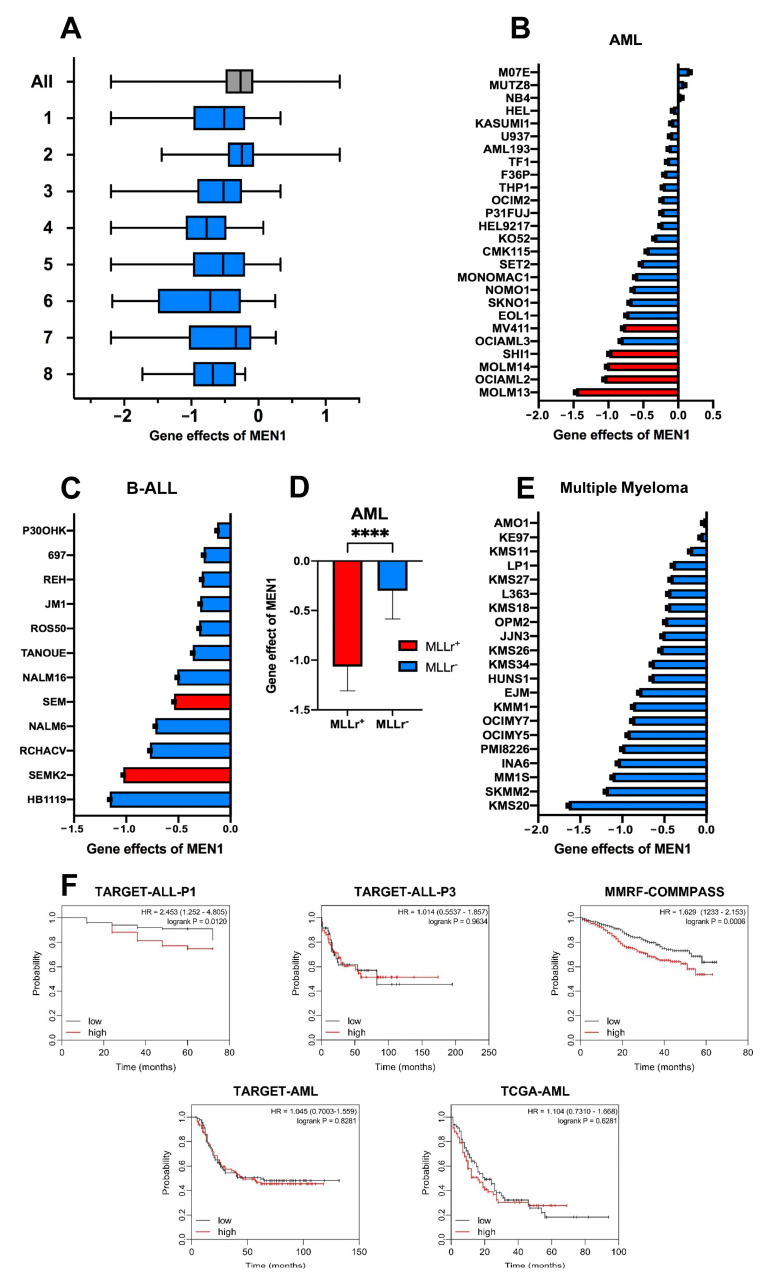
MEN1 dependency is not unique to neoplasms harboring MLLr. (**A**) MEN1 dependency is common in neoplasms of different origins. The enriched pre-existing categories (“lineages”) with *p* < 0.0005 are shown. N indicates the number of cell lines plotted in each lineage. The ranking is done based on effect size in ascending order. 1. Hematopoietic and lymphoid (N = 116); 2. Solid (N = 979); 3. Lymphoid (N = 79); 4. Plasma Cell Myeloma (N = 20); 5. Non-Hodgkin lymphoma (N = 59); 6. Diffuse Large B-Cell Lymphoma (N = 14); 7. Myeloid (N = 37); 8. B-Lymphoblastic (N = 12). For estimation of MEN1 dependency we used the “gene effect score”. The gene effect scores were normalized globally so that the median of reference nonessential genes was equal 0 and the median of essential genes was equal -1 across cell lines. Dataset: CRISPR (DepMap 22Q4, Chronos, https://depmap.org/portal/gene/MEN1?tab=overview, accessed on 28 February 2023). Enriched lineages are shown, these lines are significant with *p* < 0.0005. “All” is shown in gray, the specific lineages are shown in blue). (**B**,**C**) MEN1-dependency of individual AML (**B**) and B-ALL (**C**) cell lines. MLL-rearranged cell lines (MLLr) are marked in red all other driver mutations are marked in blue. (**D**). MLLr AML cell lines are more dependent on MEN1-expression. Gene effect values of the AML cell lines driven with MLLr (red) and other oncogenic mutations (blue) shown in (**B**) were compared using Mann-Whitney-test (t < 0.0001 (****)). (**E**) Dependency of individual MM cell lines on MEN1. (**F**) Kaplan-Meier analysis of the association of overall survival with MEN1 expression in available B-ALL, AML, and MM cohorts. Data generated by the Therapeutically Applicable Research to Generate Effective Treatments (TARGET) (https://www.cancer.gov/ccg/research/genome-sequencing/target, accessed on 23 September 2023) initiative was mined using UCSC Xena (https://xenabrowser.net/, accessed on 20 September 2023). Study accession: phs000218, phs000465, phs000748. Statistical analysis was performed with Log-rank (Mantel-Cox) test, hazard ratio was calculated using the logrank method.

**Figure 2 ijms-24-16472-f002:**
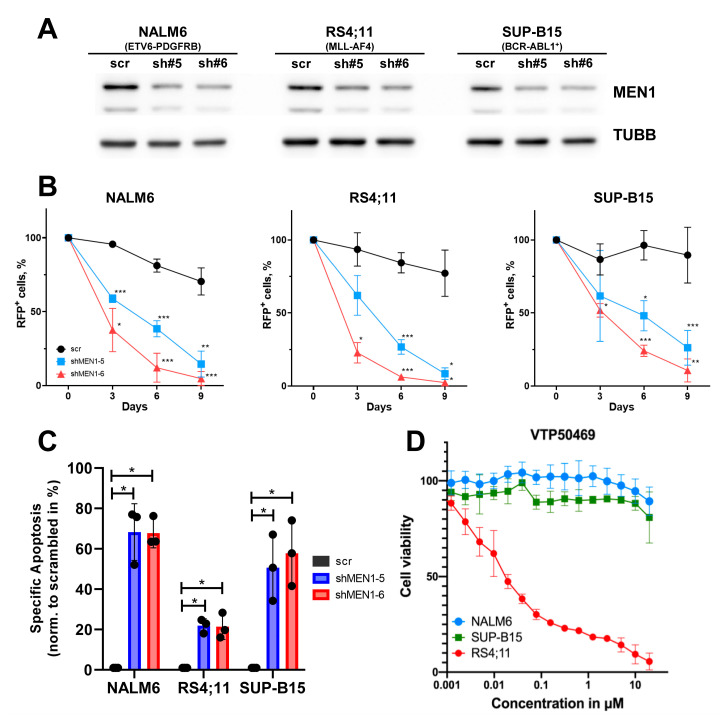
ShRNA-dependent knockdown of MEN1 decreases fitness and induces apoptosis in B-ALL cell lines independently of the presence of MLLr. (**A**–**C**) B-ALL cell lines were transduced with shRNAs targeting MEN1 (shRNA#5 or shRNA#6) or control vectors (scr) expressing the fluorescent marker RFP. (**A**) On day 4, RFP^+^ cells were sorted and MEN1 expression was measured by immunoblot. TUBB was used as loading control. (**B**) MEN1 knockdown decreases fitness of B-ALL cell lines. Starting from day 4 after transduction (day 0) the percentage of the RFP^+^ cells was measured by flow cytometry. The data are shown as percentage of RFP^+^ cells (mean ± SD N = 3). * = *p* < 0.05, ** = *p* < 0.01, *** = *p* < 0.005 (**C**) MEN1 knockdown induces apoptosis in B-ALL cell lines. RFP^+^ cells were sorted on day 3 after transduction and incubated under the normal conditions for 72 h. Apoptosis was measured by Annexin V/7-AAD staining. The data are shown as specific apoptosis (SA). SA was calculated as: SA%, =100 × (A_E_ − A_C_)/(100 − A_C_), where A_E_ and A_C_ equals the percentage of apoptotic cells in the experimental and control groups, respectively (mean ± SD N = 3). (**D**) NALM6 and SUP-B15 cell lines are not sensitive to VTP50469. The sensitivity was measured by 6 days MTT metabolic test. The data are shown as percentage to control treated with vehicle DMSO (mean ± SD, N = 3).

**Figure 3 ijms-24-16472-f003:**
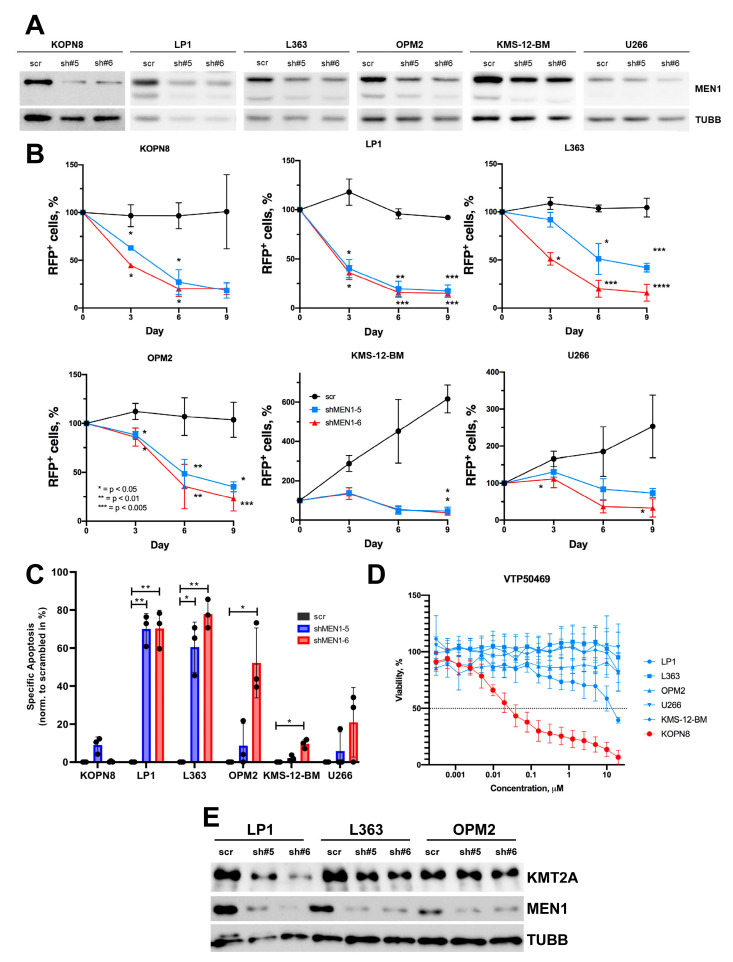
ShRNA-dependent MEN1 knockdown decreases fitness and induces apoptosis in MM cell lines (**A**–**C**) MM cell lines were transduced with shRNAs (sh#5 or sh#6) or control RFP^+^. (**A**) On day 4, RFP^+^ cells were sorted and MEN1 expression was measured by immunoblot. TUBB was used as loading control. (**B**) MEN1 knockdown decreases fitness of MM cell lines. Starting from day 4 after transduction (day 0) the percentage of the RFP^+^ cells was measured by flow cytometry. The data are shown as percent of RFP^+^ cells (mean ± SD, N = 3). (**C**) MEN1 knockdown induces apoptosis in MM cell lines. RFP^+^ cells were sorted on day 3 after transduction and incubated under normal conditions for 72 h. Apoptosis was measured by Annexin V/7-AAD staining. The data are shown as specific apoptosis as described in the legend to Figure 2. * = *p* < 0.05, ** = *p* < 0.01, *** = *p* < 0.005, **** = *p* < 0.001 (**D**) MM cell lines are not sensitive to VTP50469. The sensitivity was measured by 6 days MTT metabolic test. The MLLr B-ALL cell line KOPN8 was used as a positive control. The data are shown as percentage to control treated with vehicle DMSO (mean ± SD, N = 3). (**E**) MEN1 KD downregulates KMT2A expression in LP1 and L363 cell lines. The transduced cells were sorted and analyzed by immunoblot as described in (**A**–**C**). The representative one of three independent experiments yielding similar results is shown.

**Figure 4 ijms-24-16472-f004:**
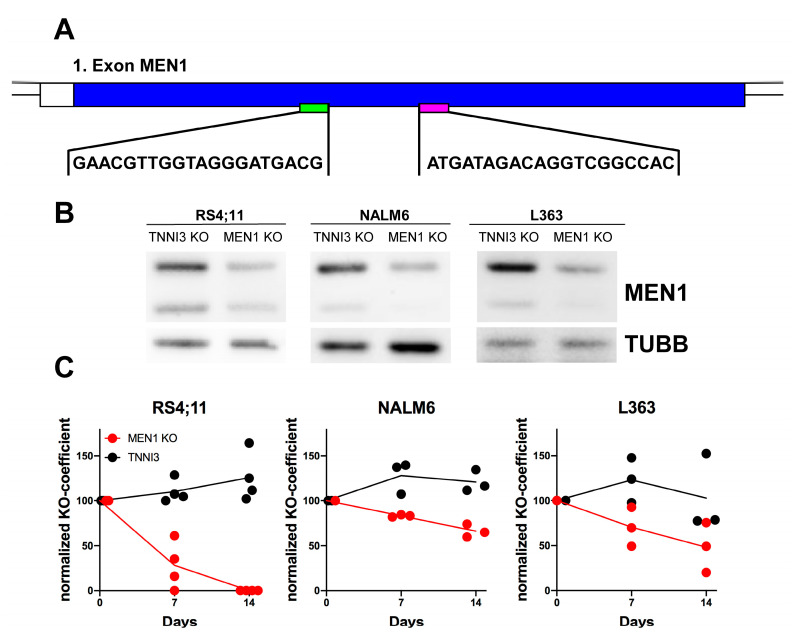
CRISPR/Cas9 editing of MEN1 decreases the fitness of MM and B-ALL cells harboring edited alleles. (**A**–**C**) NALM6 and L363 cells were transfected with RNP complexes containing tandem crRNAs targeting the 1st exon of MEN1 or control tandem crRNAs targeting the 3rd intron of TNNI3. (**A**) The target sequences in the first exon of MEN1 are shown; the pictures were generated with GenePalette (Version 2.1.1, University of California). (**B**) MEN1 protein expression one week after KO was measured by immunoblot. (**C**) The proportion of alleles harboring inactivating mutations (frameshift or indel exceeding 21 bp) is shown as a knockout coefficient (KO coefficient) normalized to the starting KO values. The statistical significance of the continuous decrease in the proportion of the MEN1 KO alleles was analyzed by linear regression analysis (L363: slope = −3.697; R^2^ = 0.6161; *p* = 0.0122; NALM6: slope = −2.412; R^2^ = 0.9406; *p* < 0.0001). The proportion of TNNI3 KO alleles slightly increased (L363: slope = 0.2004; R^2^ = 0.0020; *p* = 0.9089; NALM6: slope = 1.487; R^2^ = 0.2941; *p* = 0.1314). Data obtained from 3–4 independent experiments are shown.

## Data Availability

The data presented in this study are included in the article and in the Appendix A.

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
