# Peer review of "Dependency of B-Cell Acute Lymphoblastic Leukemia and Multiple Myeloma Cell Lines on MEN1 Extends beyond MEN1–KMT2A Interaction"

_ijms, 2023, doi:10.3390/ijms242216472_

Round 1

Reviewer 1 Report (Previous Reviewer 1)

Comments and Suggestions for Authors It is an ongoing challenge in the hematology malignancy field to find better therapies to treat AML, as HSCs acquire driver mutations that undergo progressive epigenetic changes leading to leukemia transformation. Understanding the epigenetic changes that occur and what driver mutations lead to poor prognosis and drug resistance in these patients. The study "Dependency of B-cell Acute Lymphoblastic Leukemia and multiple myeloma cell lines on MEN1 is more than ME1-KMT2A interaction" tries to add a piece to the puzzle, showing that the interaction between MEN1 and KMT2a is not limited to KMT2a but other cell lines were dependent and sensitive to MEN1 inhibition by the drug VTP50469.   Some studies published earlier this year show that treatment with revumenib leads to mutations in MEN1 and leading to tumor relapse. (Perner, Florian, et al. "MEN1 mutations mediate clinical resistance to menin inhibition." Nature 615.7954 (2023): 913-919.) This could support the manuscript showing that MEN1 correlation with other mutations and newly acquired mutations in MEN1 after inhibition can lead to drug resistance.    I appreciate the authors' efforts in making the changes in the first manuscript and the new version is well improved with all the changes requested. I have no further changes or questions.      Comments on the Quality of English Language

Although the authors did improve the original manuscripts, more could be done by this study. I understand though, the limitations the group may have and the limitations of the study itself. I think the manuscript can be accepted and has new information for those working on MEN1-Kmt2a mutations. 

Author Response

Reviewer 2 Report (Previous Reviewer 2)

Comments and Suggestions for Authors

This is a revised and resubmitted manuscropt. Much improvement has been made and I would have a few minor points.

Please enhance the image resolution. Use vectorized figures for example.

Figure 1A is exactly the same as the depmap website. Please be sure that authors have been properly authorized to re-use it.

In line 102, the citation [17] is irrelevant to Achilles project or depmap.

In line 16, "appeared efficient" is confusing. Are authors means that MEN1 binders showed some efficacy or effectiveness against MLLr AML and B-ALL in animal models and clinical studies?

It appears that VTP50469 sensitive cell lines (RS4;11, KOPN8) showed less apoptosis induction upon shMEN treatment, but still had impaired fitness. Could authors comment on to what extent the fitness may be associated with apoptosis, or other cytotoxic effects?

Author Response

This manuscript is a resubmission of an earlier submission. The following is a list of the peer review reports and author responses from that submission.

Round 1

Reviewer 1 Report

Comments and Suggestions for Authors

The paper “Dependency of B-cell Acute Lymphoblastic Leukemia and Multiple Myeloma Cell Lines on MEN1 is more than MEN-KMT2A Interaction’ analyzed several B-lymphoblastic and multiple myeloma cell lines knocking out the gene MEN1 to test the dependency of MEN1 with KMT2A gene mutation in mixed-lineage leukemia (MLLr). They hypothesized that different mechanisms than MEN1-KMT2A are instrumental to MEN1 dependency.

They used three B-ALL different cell lines: NALM6, SUP-B15, and RS4;11 (only one containing KMT2A mutation), and also the multiple myeloma cell lines: KOPN8 (Only one containing KMT2A mutation), LP1, L363, OPM2, KMS-12-BM and U266 (Not containing KMT2a mutation, but containing TP53 mutation) and used shRNA to knock out the MEN1 expression in these cell lines.

They conclude by saying that the role of MEN1 is not limited to KMT2A interaction.

The manuscript is clear and is presented in a well-structured manner. The authors used pertinent and recent publications in their references. The manuscript is also scientifically sound with the appropriate and reproducible experimental design.

I have minor concerns about the manuscript and I believe the authors will be able to address them:

Please have included in the Methods the criteria used to select the data used in Figure 1. The Methods don’t contain any information about how the Data was selected and what statistical analyses were made.

In Figure 1 D, the figure says: MLLr+ and MLLr-. It was confusing to understand what it means. Are these MLLr+ and MLLr- the cell lines used in Figure 1B-C?

In Figure 2, figures 2C and 2D are missing the control. It is not clear if the statistical analysis is made between shMEN1-5 and shMEN1-6, or the control (Not shown in the graphs.) It would be good to have the statistical method used in the figure legend (ANOVA, etc).

Figure 3: same for Figure 2. Please include the control in Figures 3C and 3D. Also in the text, it says that all cell lines are not sensitive to the VTP50469, but KOPN8 (in red 3D), it shows a decrease in viability.

For Figure 4: I would suggest doing more experiments using the CRISP/CAS9 cells. Authors are claiming that the conclusion: "MEN1 confers a selective disadvantage in B-ALL and MM cell lines which do not harbor MLLr, or other mutations conferring dependency on MEN1", but the experiment used (Figure 4C) does not corroborate this conclusion (It is a KO coefficient assay, not proliferative assay). Authors need to provide other experiments for cell proliferation to convince this conclusion. A colonies assay or another cell proliferative assay, or functional assays would be desirable. As well, if possible, use other cell lines as well, because only RS4;11 contains the KMT2A mutation.

Authors should also read the manuscript carefully: Lines 198-200 should be deleted as it doesn’t belong to the text.

Reviewer 2 Report

Comments and Suggestions for Authors

The current manuscript made a proof-of-principle finding that the fitness of multiple B-cell ALL and MM cell lines depends on the expression of  MEN1, and could be attractive to researchers in this area. However the study reported in current manuscript is too preliminary at this point. Additional experiments are recommended for publication.

Figures 2~4 are basically the same thing and one or more further steps can be taken to make the manuscript more relevant. For example, to test the effects of knock-down or knock-out in one or two mouse tumor models, or meta-analysis of patients data.

It is stated that "MEN1 depletion induces apoptosis in MM cell lines in a KMT2A-independent 157 manner". This can't be concluded from current data. Could authors please test the effects of shMEN1 on apoptosis in the presence of VTP50469, or other KMT2A inhibitors to confirm this? Does shMEN1 treatment affect KMT2A expression?

For fitness measurement, "first measurement was performed 4 - 5 days post transduction and percentage of transduced cells was set as 100% and used for normalization of results of following measurements". What is the baseline percentage of transduced (RFP+) cells on day 4~5 post transducing? Will the baseline percentage, if different among control and shRNAi groups, cause a bias in the following measurement? Could authors please show a proliferation curve of tumor cells, ie. did the compromised fitness of transduced cells also delay gross proliferation?

Minor point:

For all Western blot images, in particular the full gell image in supplemental figures, please mark the right band of each protein as well as molecular weight.
